# Displacement Reaction-Assisted Synthesis of Sub-Nanometer Pt/Bi Boost Methanol-Tolerant Fuel Cells

**DOI:** 10.3390/nano12081301

**Published:** 2022-04-11

**Authors:** Xianling Wu, Dumei Wang, Xueming Kang, Dongtang Zhang, Yong Yan, Guangsheng Guo, Zaicheng Sun, Xiayan Wang

**Affiliations:** 1Center of Excellence for Environmental Safety and Biological Effects, Beijing Key Laboratory for Green Catalysis and Separation, Department of Chemistry and Biology, Beijing University of Technology, Beijing 100124, China; wuxl@emails.bjut.edu.cn (X.W.); wangdm@emails.bjut.edu.cn (D.W.); kxm@emails.bjut.edu.cn (X.K.); yong.yan@bjut.edu.cn (Y.Y.); gsguo@bjut.edu.cn (G.G.); xiayanwang@bjut.edu.cn (X.W.); 2Minzu University of China, Beijing 100081, China

**Keywords:** cluster, electrocatalysis, electrochemistry, oxygen reduction reaction, fuel cell

## Abstract

The development of new synthetic methods for methanol-tolerant catalysts with improved performance is of fundamental importance for the commercialization of fuel cells. Herein, we reported a facile displacement reaction-assisted synthesis of graphene-supported sub-nanometer Pt/Bi catalysts (Pt/Bi/rGO). Bismuth (0) nanoparticles produced by NH_3_BH_3_ reduction can be further dissolved into the ethylene glycol, implying Bi(0) has a strong interaction with the hydroxyl group. That is the key interaction between Bi(0) and the functional group on the rGO to form the ultra-small Bi/rGO catalyst. Furthermore, Pt clusters are obtained by the displacement between Bi(0) and HPtCl_4_ and are directly anchored to the rGO surface. The as-synthesized Pt/Bi/rGO catalyst exhibits high oxygen reduction mass activity and high tolerance to methanol poisoning. In the presence of 0.5 mol/L CH_3_OH, the initial potential and activity of ORR were almost unchanged, which demonstrated great potential in the application of direct methanol fuel cells.

## 1. Introduction

The development of environmentally friendly energy sources is a challenging task. Over the past 20 years, direct methanol fuel cells (DMFCs) have attracted increasing attention from scientists for their high energy-conversion efficiency and energy density, low operating temperature, and low-to-zero pollutant emission [1,2,3]. Currently, platinum is widely used for DMFCs, as both the cathode and anode, because of its high catalytic activity and excellent stability [4]. Although DMFCs, in theory, have a very high energy density and good commercial potential, their cathode catalytic activity is low, Pt has a high cost as the cathode oxygen reduction catalyst, and “methanol permeation” has always restricted the development of commercial DMFCs [5]. The methanol penetrates the proton membrane to the cathode, causing simultaneous methanol oxidation and oxygen reduction reactions to generate a mixed potential, which dramatically reduces the overall output voltage. Moreover, methanol also poisons Pt catalysts, significantly reducing their activity and durability. The “methanol permeation” becomes aggravated with an increase in the methanol concentration, which impedes achieving high-power DMFCs by using methanol with high concentrations. Consequently, many scientists have focused on the study of methanol-tolerant catalysts and great progress has been made [6,7,8].

For solving the problem of “methanol penetration”, two aspects can be considered: (i) designing an advanced proton exchange membrane [9,10]; (ii) preparing a high-performance methanol-tolerant cathode catalyst [11,12]. At present, the Nafion membrane is of the widely used proton exchange membranes for DMFCs, and the methanol molecules can easily cross over the Nafion membrane. Therefore, on the premise of ensuring ion migration, the design of a new proton membrane is one of the solutions to the problem of methanol penetration. Another way to solve the problem is to design an advanced catalyst with a high oxygen reduction reaction and methanol-tolerant performance, ensuring that the DMFC system can still work efficiently under the penetration of a certain concentration of methanol. Some methods, such as Platinum-modified [13], carbon defect-anchored Pt single-atom [14,15,16] and Pt-based alloy [17,18], have verified that the methanol-tolerant nature of the catalyst can be improved. PtBi exhibits high tolerance to methanol and co poisoning, but the activity and stability are still limited due to the large particle sizes and aggregation behavior during the catalytic process [19,20]. The preparation of ultra-small and stable Pt-based catalysts in the liquid phase faces great challenges from the migration and aggregation of active Pt atoms [16,21]. The use of support is an effective way to improve the dispersion and stability of catalysts [22] and the carbon material graphene has enjoyed extensive use as a substrate for high dispersion catalysts, owing to its unique geometry and electronic properties [23,24].

In this study, an attempt was made to synthesize ultra-small Pt/Bi catalyst support on reduced graphene oxide (rGO), for a highly active and stable oxygen reduction reaction with methanol tolerance. First, Bi is reduced and loaded on the surface of rGO. The excess Bi in the solution can completely consume the reducing agent, and then the crystal structure of Bi is dissolved in the solution over 1 h, whereas Bi atoms on the carrier are stabilized by the riveting effect of graphite. Then, an H_2_PtCl_6_ solution is added, and the larger electrode reduction potential between Bi and Pt promotes Pt to replace Bi with zero valences, finally forming an ultra-small Pt/Bi/rGO catalyst. We found that the prepared catalyst had good oxygen reduction reaction (ORR) activity and methanol tolerance, which provides a new way to solve the problem of “methanol permeation”. This method is simple and feasible and provides a new idea for the preparation of single-atom catalysts in the liquid phase. In particular, the idea of using ethylene glycol to dissolve excess simple-substance Bi may also be suitable for the preparation of other metal catalysts.

## 2. Materials and Methods

### 2.1. Chemical Reagents

Chloroplatinic acid hexahydrate (H_2_PtCl_6_·6H_2_O, 99.9%), bismuth nitrate (Bi(NO_3_)_3_·5H_2_O, 99.0%), ethylene glycol (EG, 99%), polyvinylpyrrolidone (PVP, K30, 99.9%) were purchased from Sinopharm Chemical Reagent Co., Ltd. (Beijing, China). The borane–ammonia complex (H_3_N·BH_3_, 90%), commercial Pt/C (Pt/C, 20 wt%), and Nafion 117 solution (5%) were purchased from Sigma Aldrich (St. Louis, MO, USA). Ultrapure water (18.2 MΩ cm) was used in the experiment.

### 2.2. Preparation of Pt/Bi/rGO DACs Catalyst

First, 10 mg rGO, 48.5 mg Bi(NO_3_)_3_·5H_2_O and 167 mg PVP were dissolved in 50 mL EG confined in a beaker under constant magnetic stirring. Graphene was added and well distributed using an ultrasonic cleaning bath and 6 mg NH_3_BH_3_ was then added into the solution, after an hour H_2_PtCl_6_ was then added. After that, the solution and the mixture were continually stirred for 20 h. The final products were collected by a centrifugation process and then washed successively with ethanol. As identified below, the produced catalyst at the single atomic level was obtained. Because of the addition of graphene to form a supported catalyst, the catalyst should be expressed as Pt/Bi/graphene.

### 2.3. Characterization

The X-ray diffraction (XRD) patterns for the samples were obtained using a Bruker D8 Advance diffractometer with Cu-Kα (λ = 1.5405 Å) radiation source (40 kV, 40 mA). Transmission Electron Microscopy (TEM, JEOL, Tokyo, Japan) was carried out with a JEOL JEM-2100 microscopy (JEOL, Tokyo, Japan) operating at 200 kV with a nominal resolution, and high-resolution scanning transmission electron microscopy (HRTEM, JEOL, Tokyo, Japan) was performed on JEOL ARM200F. The composition of the prepared catalysts was measured using an inductively coupled plasma atomic emission spectrometer (ICP-AES, PerkinElmer, Waltham, MA, USA) on an IRIS Intrepid spectrometer after the dissolution of the samples in aqua regia.

### 2.4. Electrochemical Measurement

The electrochemical measurements were performed on a CHI 1030C at room temperature. The glassy carbon rotating disk electrode (GC-RDE, 4 mm, BAS, West Lafayette, IND, USA) was used as the working electrode. A GC film electrode was used as the counter electrode. All the potentials in this study are presented with reference to Ag/AgCl. Before using the GC electrode as a substrate for the catalysts, it was polished with 0.05 mm alumina to yield a mirror finish. Cyclic voltammetry (CV) measurements were carried out in N_2_-saturated 0.5 mol/L H_2_SO_4_ solutions at 50 mV/s. The ORR measurements were performed in O_2_-saturated 0.5 mol/L H_2_SO_4_ solutions using GC-RDE at a sweep rate of 10 mV/s. In the ORR polarization curve, the current densities were normalized with reference to the Pt mass loading on the GC-RDE. The potential vs. Ag/AgCl reported in the manuscript was converted to the RHE according to the following equation:Potential (V vs. RHE) = Applied potential (V vs. Ag/AgCl/sat. KCl) + 0.199 V + 0.0592 × pH.

## 3. Results

The schematic representation of the displacement reaction route used to synthesize the ultra-small Pt/Bi on rGO (Pt/Bi/rGO) catalysts is shown in Figure 1. First, the rGO was dispersed in Bi(NO_3_)_3_/ethylene glycol solution, and the abundance of oxygen-containing functional groups on the rGO provided active sites for the chemical adsorption of Bi^3+^ ions. Subsequently, NH_3_BH_3_ was added to the reaction as the reductant to form Bi in the rGO dispersion (Figure 1a). Second, by stirring the solution for another 1 h, the Bi particles formed in the solution were dissolved, leaving only the Bi particles stabilized by the rGO (Figure 1b). Lastly, H_2_PtCl_6_ was added and stirred for 20 h to fully react and obtain the ultra-small Pt/Bi/C (Figure 1c).

To demonstrate that the Bi(0) nanoparticles can be redissolved in the ethylene glycol, the control experiment was carried out (Figure 2 and Appendix A). Figure 2A presents Bi(NO_3_)_3_ solution in the ethylene glycol, with a colorless and transparent solution. After NH_3_BH_3_ was added, the solution changed from colorless to black (Figure 2B). After continuous stirring for approximately 45 min, the darkened solution became colorless again, which implied the resulting Bi(0) particles had redissolved into the solution. Further, no solid products appeared after the centrifugation and filtration of the solution. With the presence of rGO, the initial black product (after reaction for 2 min, Bi/rGO-2 min) was collected by centrifugation and the X-ray diffraction (XRD) patterns are presented in Figure 3A. The peaks at 27.2°, 37.9°, 39.6°, and 48.7° can be indexed to the (012), (104), (110), and (202) reflections of simple-substance hexagonal Bi (R-3m), respectively. This indicated that Bi^3+^ could be reduced to Bi^0^ by NH_3_BH_3_. Further, the solid product was re-dispersed into ethylene glycol and stirred for over an hour. The Bi^0^ nanoparticles can be dissolved in the ethylene glycol according to Figure 2. XRD results (Figure 3A) of the products (Bi/rGO-1 h) show no obvious diffraction peaks, implying that the Bi nanoparticles on the rGO redissolve in the ethylene glycol. ICP-AES was employed to detect the Bi amount on the rGO surface. The results show that the content of the Bi element in the sample is approximately 6%. This implied that Bi was adsorbed on the surface of rGO because the rGO surface contains abundant -OH and -COOH groups, which are helpful to anchor the Bi^0^ atoms. The dissolution of bismuth in ethylene glycol ensures the mono-dispersity of Bi (or near-atomic scale) on the rGO surface.

Furthermore, after the addition of H_2_PtCl_6_, the Bi^0^ atoms on the surface of rGO were displaced to Pt^0^. The displacement reaction was between Bi(0) and H_2_PtCl_4_. The displacement reaction only occurs at the Bi sites on the surface of rGO, which effectively avoids the formation of large particles. The displacement reaction on the surface of the rGO directly anchors Pt^0^ atoms to the rGO, which increases the dispersion and stability. The XRD result of the final Pt/Bi/rGO showed no obvious diffraction peak (Figure 3A), indicating that no Pt nanoparticles formed. The ICP-AES results showed both the existence of Pt and Bi in the final product. X-ray photoelectron spectroscopy was used to disclose the composition and valence state. The XPS survey showed that Bi, Pt C, and O exist in the sample (Appendix A XPS). The high-resolution XPS spectra are shown in Figure 3B. The typical three peaks are attributed to Pt 4f_7/2_ at 71.1 eV, 72.4 eV, and 74.1 eV, respectively. Each peak can be deconvoluted into three components, which are assigned to Pt^0^, Pt^2+^, and Pt^4+^. In addition, the Bi signal is also observed at 159.3 eV, which is assigned to Bi 4f_7/2_ (Appendix A). Because Bismuth is easily oxidized in the air atmosphere, the results show that Bismuth is in an oxidation state.

The dispersion and size of Pt on the surface of graphene were further characterized by TEM. The TEM images of the bright-field and dark-field of the prepared highly dispersed catalyst are shown in Appendix A. There were no obvious particles observed through TEM and HRTEM images, whereas the EDS results indicated the presence of Pt and Bi elements (Appendix A). The Pt element loading was determined to be 3.5% by ICP. Obvious bright spots could be seen in the HAADF images because the atomic mass of Pt and Bi are greater than that of C, which improves the contrast. The spots were uniformly dispersed on the surface of the rGO without aggregation, and all the observed particles were less than 1 nm. To better understand the distribution of Pt and Bi elements, the EDS mapping analysis was performed. As shown in Figure 4, Pt and Bi were evenly dispersed throughout the graphene. This result indicates that the dispersion of Pt and Bi elements on the surface of the rGO was homogeneous.

For comparison, Pt/rGO was prepared under the same conditions, except for the absence of Bi(NO_3_)_3_. As shown in the TEM results in Appendix A, in the absence of Bi, Pt particles with a diameter of 2.9 nm appeared in the product, and the atomically dispersed product could not be obtained. We concluded that Bi^3+^ could be reduced to Bi^0^ by NH_3_BH_3_ and that Bi crystals could be dissolved in ethylene glycol. In contrast, Bi^0^ atoms on the surface of graphene are insoluble owing to anchor stress. PtCl_6_^2−^ can react with Bi^0^ on the rGO surface; the resulting products are Bi^3+^ and Pt^0^. Pt^0^ is directly anchored to the surface of the rGO instead of growing in solution to form particles. Therefore, the atomic Pt/Bi on the carbon catalyst is prepared by the spontaneous galvanic displacement reaction on the rGO surface, driven by the difference in electrochemical potential between Bi and Pt. This is different from the conventional liquid-phase reduction reaction. On account of the action of the reducing agent, when metal ions are reduced to atoms in conventional liquid phase reactions, they can easily nucleate, grow, and form nanoparticles, owing to the large free energy. Herein, the displacement reaction between atomically dispersed Bi/rGO and PtCl_6_^2−^ not only ensures the atomic dispersion of the Pt0, but also improves the dispersion on the surface of the rGO.

The methanol-tolerant performance of Pt/Bi/rGO was investigated electrochemically by modifying a glassy carbon electrode with ultra-small Pt/Bi/rGO and Pt/C. The cyclic voltammetry (CV) curves of the modified electrodes in aqueous solutions of H_2_SO_4_ (0.5 mol/L) and CH_3_OH (0.5 mol/L) at room temperature are shown in Figure 5A and Appendix A. From the results, it is clear that in the scanning range of 0–1.2 V, two peaks appear in the positive sweep, at 0.85 V and 0.6 V, respectively, which are the typical methanol oxidation peaks; namely, the adsorption oxidation peaks of methanol and the adsorption oxidation peaks of intermediate products. However, the prepared ultra-small Pt/Bi/rGO catalyst had no obvious catalytic oxidation effect after methanol was added, indicating that the synthesized ultra-small Pt/Bi/rGO catalyst was inert to methanol. This may be because Pt atoms are highly dispersed on the carbon support, preventing the formation of Pt–Pt bonds, which hinder the adsorption and dissociation of methanol molecules on the catalyst, thus, providing resistance to methanol oxidation. As revealed by the linear sweep voltammetry (LSV) of ultra-small Pt/Bi/rGO at different rotation speeds under the oxygen saturation condition of 0.5 mol/L H_2_SO_4_, the current gradually increases as rotation speed increases, indicating that the obtained single-atom catalyst had notable oxygen reduction performance (Appendix A). In addition, rGO was also tested by LSV under the same conditions. It was found that the current of rGO did not change significantly with an increase in speed when the rGO was above 0.2 V, which indicates that pure rGO has no catalytic performance for oxygen reduction (Appendix A). The LSV performance of rGO, commercial Pt/C, and the as-prepared ultra-small Pt/Bi/rGO catalyst was evaluated in the 0.5 mol/L O_2_-saturated H_2_SO_4_; ultra-small Pt/Bi/rGO exhibited much higher mass activity than Pt/C (Appendix A). Further, the LSV test of commercial Pt/C (red) and Pt/Bi/rGO (black) was conducted in solutions containing 0.5 mol/L H_2_SO_4_ (dashed line) and 0.5 mol/L H_2_SO_4_ + 0.5 mol/L methanol aqueous solution (solid line), respectively (Figure 5B). It is clear that the onset potential and mass activity current of commercial Pt/C changes significantly in the presence of methanol, whereas the corresponding values for ultra-small Pt/Bi/rGO remained nearly stable. At a current density of 60 mA mg^−1^_Pt_, the potential was decreased from 0.79 V to 0.51 V, whereas the potential of Pt/Bi showed a very small difference. These tests indicate that the ultra-small Pt/Bi/rGO has excellent CH_3_OH tolerance. The resistance of the ultra-small Pt/Bi/rGO catalyst to poisoning can significantly promote the development of DMFC catalysts.

## 4. Conclusions

In conclusion, we have developed a displacement method to prepare sub-nanometer Pt/Bi/rGO for high ORR and methanol tolerance performance. The nanocrystalline Bi^0^, prepared by reduction, can be dissolved in ethylene glycol, which provides a basis for the preparation of atomically dispersed catalysts. Owing to the anchoring effect on the carrier, the near atomically dispersed metal catalyst can be formed and stabilized. We found that the as-prepared catalyst showed almost no methanol oxidation activity, which provides a new avenue to solve the problem of “methanol permeation”. The single-atom preparation method is simple and feasible and provides a new idea in the preparation of high-dispersion catalysts. In particular, the use of ethylene glycol to dissolve excess simple-substance Bi should provide a reference for the preparation of other metal catalysts.

## Figures and Tables

**Figure 1 nanomaterials-12-01301-f001:**
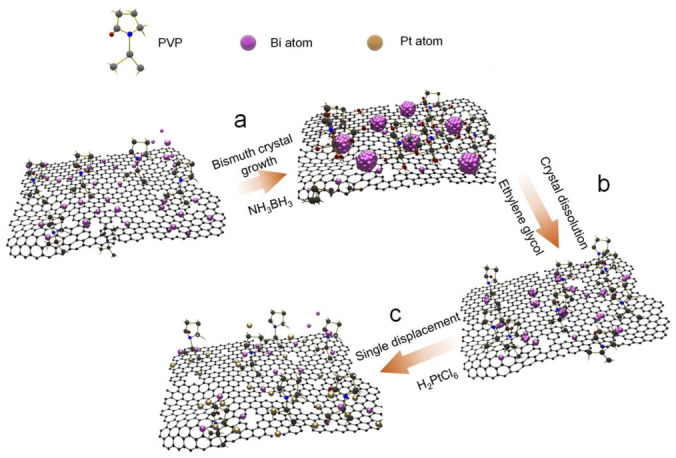
Illustrations of the facilitation and stabilization process show the displacement reaction route leading to the formation of ultra-small Pt/Bi/rGO. (**a**) Bismuth crystal grows on rGO surface, (**b**) Bi particles in solution are dissolved, (**c**) displacement reaction occurs.

**Figure 2 nanomaterials-12-01301-f002:**
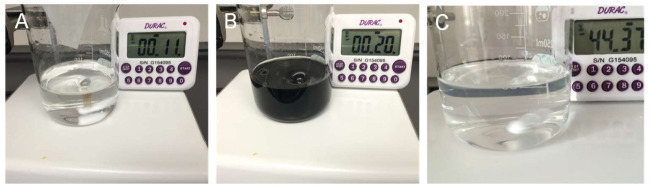
Photographs of the experimental process of Bi(NO_3_)_3_ reduction with NH_3_BH_3_ without rGO support.(**A**) Bi(NO_3_)_3_ solution in the ethylene glycol, (**B**) NH_3_BH_3_ added in the solution, (**C**) Continuous stirring for approximately 45 min.

**Figure 3 nanomaterials-12-01301-f003:**
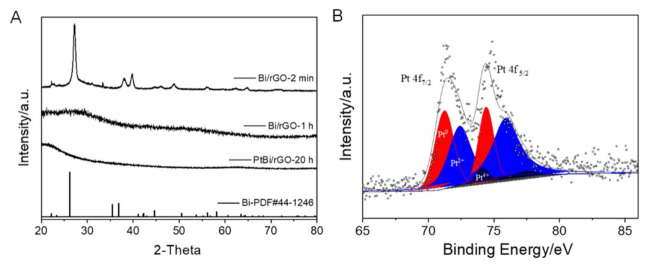
(**A**) XRD patterns of as-synthesized Bi/rGO-2 min, Bi/rGO-1 h, and Pt/Bi/rGO, (**B**) high-resolution XPS spectra of Pt in Pt/Bi/rGO.

**Figure 4 nanomaterials-12-01301-f004:**
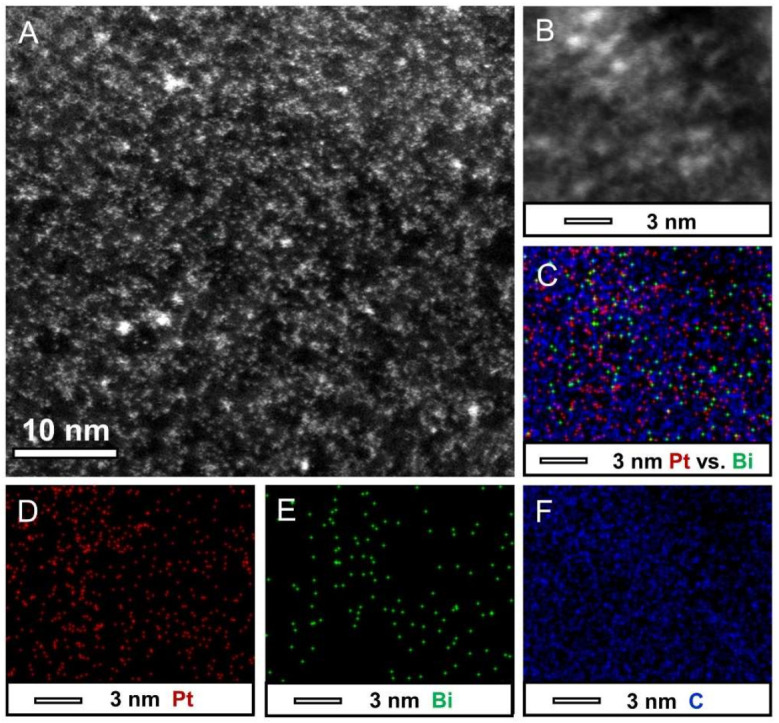
HRTEM images, HAADF-STEM, and EDS element mapping images of ultra-small Pt/Bi/rGO. (**A**) HRTEM images of ultra-small Pt/Bi/rGO, (**B**) HAADF-STEM-EDS mapping image, (**C**) overlapped image, and (**D**–**F**) HAADF-STEM-EDS elemental mapping images for Pt, Bi, and C, respectively, taken for the area in (**B**).

**Figure 5 nanomaterials-12-01301-f005:**
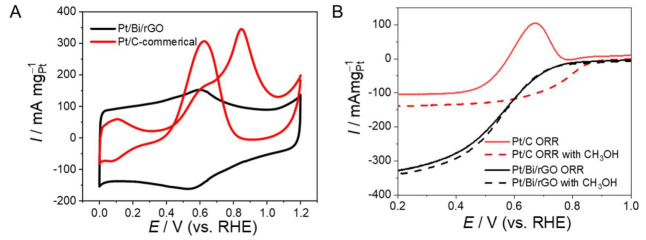
(**A**) CV curves measured in 0.5 mol/L H_2_SO_4_ and 0.5 mol/L methanol aqueous solution for the Pt/C and ultra-small Pt/Bi/rGO catalysts, respectively. (**B**) Comparison of oxygen reduction properties in 0.5 mol/L H_2_SO_4_ (dashed line) and 0.5 mol/L H_2_SO_4_ + 0.5 mol/L methanol aqueous solution (solid line). All current densities are normalized against the mass of metallic Pt loaded onto the electrodes.

## Data Availability

The data presented in this study are available upon request from the corresponding authors.

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
