# Peer review of "Displacement Reaction-Assisted Synthesis of Sub-Nanometer Pt/Bi Boost Methanol-Tolerant Fuel Cells"

_nanomaterials, 2022, doi:10.3390/nano12081301_

Round 1

Reviewer 1 Report

The article deals with a new synthetic method to prepare methanol-tolerant catalysts. The idea behind the work sounds good, however, in my opinion some points should be clarified before publication. The main concern is that many results are presented in form of supplementary figures, which however are not available. Thus the discussion can be followed only superficially. 

Other points:

  • 51-55 please revise the english language
  • 63 ...supported on graphene
  • 69 Pt to replace Bi  (or rephrase the sentence)
  • 96 microscopy
  • 97 the final part of the sentence is missing
  • 127 the sentence is not complete
  • 129 please, delete the addition of (it's a repetition)
  • 143 is it possible to show this result?
  • 138 onwards: how can you measure the XRD pattern if the solid product is dissolved in EG?
  • 152 revise the english language
  • 159 valence
  • 163 no supplementary materials was present; it might be useful to add a figure with the Bi XPS spectra, also
  • 182 some points in fig 4A seem in a similar range 2-3 nm
  • 202-203 can give more details on how the glassy carbon electrodes were modified?
  • 224 with and without methanol?
  • 228 to end: the explanation of fig. 5B is not clear, please try to improve it
  • 250-251 please update labels of suppl materials

Author Response

Point-by-point response:

Reviewer 1

The article deals with a new synthetic method to prepare methanol-tolerant catalysts. The idea behind the work sounds good, however, in my opinion some points should be clarified before publication. The main concern is that many results are presented in form of supplementary figures, which however are not available. Thus the discussion can be followed only superficially.

Response: We appreciate the reviewer for the valuable comments and questions that help us significantly improve the revised manuscript. We are sorry for the inconvenience caused by the lack of supporting materials. We have uploaded the supporting materials in the revised manuscript.

Other points:

51-55 please revise the english language

Response: We have rewritten the sentence as follows:

Some methods such as Platinum-modified,[13] carbon defect-anchored Pt single-atom,[14-16] and Pt-based alloy[17,18] have been verified that the methanol-tolerant of the catalyst can be improved.

63 ...supported on graphene

Response: We have revised it to “reduced graphene oxide (rGO)”.

69 Pt to replace Bi (or rephrase the sentence)

Response: Thanks for pointing out that. We have changed the revised manuscript to “Pt to replace Bi”

97 the final part of the sentence is missing

Response: We have rewritten the sentence as follows:

Transmission Electron Microscopy (TEM) was carried out with a JEOL JEM-2100 microscope operating at 200 kV with a nominal resolution, and high resolution scanning transmission electron microscopy (HRTEM) was performed on JEOL ARM200F.

127 the sentence is not complete

Response: We have completed the sentence in the revised draft:

Figure 1. Illustrations of the facilitation and stabilization process show the displacement reaction route leading to the formation of ultra-small Pt/Bi/rGO. (a) Bismuth crystal grows on rGO surface, (b) Bi particles in solution are dissolved, (c) Displacement reaction occurs.

129 please, delete the addition of (it's a repetition)

Response: We appreciate the reviewer for the careful review. We have deleted it.

143 is it possible to show this result?

Response: The ICP results have been described in the manuscript, the content of the Bi element in the sample is approximately 6%.

138 onwards: how can you measure the XRD pattern if the solid product is dissolved in EG?

Response: The XRD result here is the product of a reaction for 2 min, at this time the Bismuth particles have not yet dissolved. It will not be completely dissolved in 45 min.

152 revise the english language

Response: We have rewritten the sentence as follows:

Furthermore, after the addition of H2PtCl6, the Bi0 atoms on the surface of rGO are displaced to Pt0.

159 valence

Response: We have corrected it.

163 no supplementary materials was present; it might be useful to add a figure with the Bi XPS spectra, also

Response: We are sorry for the inconvenience caused by the lack of supporting materials. We have uploaded the supporting materials in the revised manuscript.

182 some points in Fig 4A seem in a similar range 2-3 nm

Response: Yes, this may be caused by partial graphene layer stacking or graphene wrinkle, or there are a large number of defects in graphene here that cause a large number of atoms to accumulate here.

202-203 can give more details on how the glassy carbon electrodes were modified?

Response: Thanks for the suggestion. More details about modifying the carbon electrodes are given in the supporting materials.

224 with and without methanol?

Response: Without methanol. In the supporting material, we describe the test conditions.

228 to end: the explanation of fig. 5B is not clear, please try to improve it

Response: Thanks for the suggestion. We have made some changes described to make it clear:

Further, the LSV test of commercial Pt/C (red) and Pt/Bi/rGO (black) was conducted in solutions containing 0.5 mol/L H2SO4 (dashed line) and 0.5 mol/L H2SO4 + 0.5 mol/L methanol aqueous solution (solid line), respectively (Figure 5B).

250-251 please update labels of supplementary materials

Response: We have uploaded the supporting materials in the revised manuscript.

Reviewer 2 Report

Wu et al. developed a displacement method to prepare sub-nanometer Pt/Bi/rGO catalyst for high ORR and methanol tolerance performance. The as-prepared catalyst showed almost no methanol oxidation activity to solve the problem of “methanol permeation. I could not judge the manuscript correctly because I did not see the supporting information file however there are several comments should be figured out, please find them here:

  1. In the abstract, the reduction of NH3BH3 is a wrong description since it is used as a reducing agent, please revise it carefully.
  2. Please write the values of onset reduction potential of ORR in numbers vs. RHE compared to the commercial Pt/C, however using the expensive Pt here is a disadvantage, why you did not use abundant transition single metal atoms, such as Co and/or Ni?
  3. In the introduction the authors highlighted the high cost and toxicity of Pt, however they used it in their catalyst.
  4. I wondered why did the authors use two reference electrodes Hg/Hg2SO4 and Ag/AgCl?
  5. Where is SI file?
  6. The XPS reflects presence of various oxidation states of Pt which is not consistent to the title and abstract.
  7. What is the exact content of Pt and Bi to carbon in your catalyst?
  8. The authors did not write anything about the catalyst long term stability and did not determine number of electrons transferred.

Author Response

Point-by-point response:

Reviewer 2

Wu et al. developed a displacement method to prepare sub-nanometer Pt/Bi/rGO catalyst for high ORR and methanol tolerance performance. The as-prepared catalyst showed almost no methanol oxidation activity to solve the problem of “methanol permeation. I could not judge the manuscript correctly because I did not see the supporting information file however there are several comments should be figured out, please find them here:

Repsonse: We appreciate the reviewer for providing helpful comments. We are sorry for the inconvenience caused by the lack of supporting materials. We have uploaded the supporting materials in the revised manuscript.

  1. In the abstract, the reduction of NH3BH3 is a wrong description since it is used as a reducing agent, please revise it carefully.

Response: Done. We have to rewrite the sentence as follows:

Bismuth (0) nanoparticles produced by NH3BH3 reduction can be further dissolved into the ethylene glycol, implying Bi(0) has a strong interaction with the hydroxyl group.

  1. Please write the values of onset reduction potential of ORR in numbers vs. RHE compared to the commercial Pt/C, however using the expensive Pt here is a disadvantage, why you did not use abundant transition single metal atoms, such as Co and/or Ni?

Response: In this manuscript we showed the ORR performance in 0.5 mol/L H2SO4 with and without 0.5 mol/L methanol. It is difficult to determine the initial potential in the presence of methanol due to Pt having good catalytic oxidation activity against methanol. Here we compared the changes in half-wave potential (current density of 60 mA mg-1Pt) with and without methanol.

We agree with the review, however, currently, Pt is still the most widely used for DMFCs as both the cathode and anode because of its high catalytic activity and excellent stability. Therefore, our goal is to solve the problems of Pt catalyst dispersion on supported and product methanol-tolerant catalysts. The standard electro potential of Co and Ni is -0.28 and -0.257, respectively, which is lower than that of Bi (+0.32). Thus, this method cannot achieve Co2+ or Ni2+ to zero valence states.

  1. In the introduction the authors highlighted the high cost and toxicity of Pt, however they used it in their catalyst.

Response: Yes, Pt nanocrystals catalyst is suffering from the high cost and “methanol penetration”. It is undeniable that Pt is still the most widely used for DMFCs as both the cathode and anode. Herein, we attempt to solve the problems by reducing the Pt load amount and increasing the methanol-tolerant.

  1. I wondered why did the authors use two reference electrodes Hg/Hg2SO4 and Ag/AgCl?

Response: Thanks to the reviewer to point this out. In this manuscript, we used Ag/AgCl as the reference and we have revised the problem.

  1. Where is SI file?

Response: We are sorry for the inconvenience caused by the lack of supporting materials. We have uploaded the supporting materials in the revised manuscript.

  1. The XPS reflects presence of various oxidation states of Pt which is not consistent to the title and abstract.

Response: Although XPS reflects various oxidation states of Pt, this is not contradictory to our conclusion. In the samples we prepared, the Pt size is very small and highly dispersed, and the XPS test requires the sample to be dry and exposed to air. In this process, partial oxidation is inevitable, which can be the reason for the various oxidation states of Pt.

  1. What is the exact content of Pt and Bi to carbon in your catalyst?

Response: The final samples were characterized by ICP and the Pt element loading was determined to be 3.5%.

  1. The authors did not write anything about the catalyst long term stability and did not determine number of electrons transferred.

Response: In this manuscript, we want to show a novel preparation method of Pt/Bi/rGO with high dispersion and methanol-tolerant. We provided the detailed preparation process and reaction mechanism, and tested the methanol-tolerant in detail. According to the previous reports, in the presence of PVP, the electrochemical stability of Pt catalyst is greatly improved. Similar to our work, Pt Single Atom loaded on carbon nanotubes showed excellent electrochemical stability, [1] and also Bi modified Pt aupported on carbon showed excellent electrochemical stability.[2] Moreover, a study of the more electrochemical performance of the highly dispersed Pt/Bi/rGO will be presented in forthcoming publications.

[1] Li, C., Chen, Z., Yi, H., Cao, Y., Du, L., Hu, Y., Kong, F., Kramer Campen, R., Gao, Y., Du, C., Yin, G., Zhang, I. Y., Tong, Y., Polyvinylpyrrolidone-Coordinated Single-Site Platinum Catalyst Exhibits High Activity for Hydrogen Evolution Reaction. Angew. Chem. Int. Ed. 2020, 59, 15902-15907.

[2] Choi, Mihwa, Ahn, Chi-Yeong, Lee, Hyunjoon, Kim, Jong Kwan, Oh, Seung-Hyeon, Hwang, Wonchan, Yang, Seugran, Kim, Jungsuk, Kim, Ok-Hee, Choi, Insoo, Sung, Yung-Eun, Cho, Yong-Hun, Rhee, Choong Kyun, Shin, Woonsup, Bi-modified Pt supported on carbon black as electro-oxidation catalyst for 300 W formic acid fuel cell stack. Applied Catalysis B-Environmental 2019, 253, 187-195.

Round 2

Reviewer 2 Report

I can see obviously that the authors figure out all comments accordingly and the manuscript was improved. I have not more comments on this manuscript.